# Hierarchical Aggregation for Numerical Data under Local Differential Privacy

**DOI:** 10.3390/s23031115

**Published:** 2023-01-18

**Authors:** Mingchao Hao, Wanqing Wu, Yuan Wan

**Affiliations:** 1School of Cyber Security and Computer, Hebei University, Baoding 071000, China; 2Key Laboratory of High Trusted Information System in Hebei Province, Hebei University, Baoding 071000, China

**Keywords:** local differential privacy, numerical data, hierarchical aggregation, stochastic gradient descent, linear regression

## Abstract

The proposal of local differential privacy solves the problem that the data collector must be trusted in centralized differential privacy models. The statistical analysis of numerical data under local differential privacy has been widely studied by many scholars. However, in real-world scenarios, numerical data from the same category but in different ranges frequently require different levels of privacy protection. We propose a hierarchical aggregation framework for numerical data under local differential privacy. In this framework, the privacy data in different ranges are assigned different privacy levels and then disturbed hierarchically and locally. After receiving users’ data, the aggregator perturbs the privacy data again to convert the low-level data into high-level data to increase the privacy data at each privacy level so as to improve the accuracy of the statistical analysis. Through theoretical analysis, it was proved that this framework meets the requirements of local differential privacy and that its final mean estimation result is unbiased. The proposed framework is combined with mini-batch stochastic gradient descent to complete the linear regression task. Sufficient experiments both on synthetic datasets and real datasets show that the framework has a higher accuracy than the existing methods in both mean estimation and mini-batch stochastic gradient descent experiments.

## 1. Introduction

Because of its strict mathematical definition and its ability to provide privacy guarantees that do not depend on the attacker’s background knowledge, the differential privacy model [1,2,3] has received attention and research from many scholars in the field of privacy protection ever since it was proposed. However, in the classical centralized differential privacy model, the data collector has direct access to the user’s real data, so it must be ensured that the data collector is trusted; otherwise, the user’s private data are at risk of leakage. To solve this problem, local differential privacy models [4,5] are proposed. The local differential privacy model allows the data collector to obtain the desired data characteristics through statistical analysis without directly accessing original user data [6,7]. Therefore, more and more companies are trying to use local differential privacy models to protect user privacy. For example, Apple [8], Google [9] and Microsoft [10] have used local differential privacy models to protect users’ privacy in their products and services.

Categorical and numerical data are two of the most fundamental types of data used in statistical analysis. The statistical analysis of these two types of data under local differential privacy models has been well studied in the literature. For example, for the frequency estimation of categorical data, the literature [11,12,13] has proposed some effective privacy-preserving mechanisms and statistical analysis methods, and for the mean estimation problem of numerical data, the literature [14,15,16] has proposed their solutions based on the RR [17] and Laplace [2] mechanisms, respectively.

In real-world application scenarios, numerical data with different characteristics often require different degrees of privacy protection. For example, in the case of income statistics, data in the lower and higher income ranges are often more sensitive and require stricter privacy protection than data in the middle income range. Moreover, in the case of user weight data, data in the normal weight range require more lenient privacy protection than fat and thin weight data. Under local differential privacy, a stronger degree of privacy protection often means poorer data availability. Therefore, when collecting numerical data from users, using a hierarchical collection approach to assign different privacy levels to data in the different ranges can significantly improve the overall usability of the privacy data. To achieve this need, Gu et al. [18] proposed a hierarchical collection approach for categorical data that allows different privacy levels for different categories of private data; the literature [19,20,21] has proposed personalized privacy solutions that allow different users to set their own privacy levels.

In this paper, we investigate the selection of different privacy protection levels for different ranges of privacy data in a local differential privacy model for the collection and analysis of numerical data. In this application setting, the following challenges exist to make the collection and analysis of privacy data satisfy the local differential privacy requirements: (1) The privacy level used in local perturbation reflects the value interval to which the privacy data belong, which also belong to user privacy, so the privacy level of the privacy data cannot be sent directly to the data collector. (2) The output value domains of user data under different privacy levels for local perturbation must be the same, so as to ensure that the attacker does not infer the user’s data from the data sent to the collector by the user. (3) The hierarchical collection of user data sets actually partitions the value domain of privacy data into intervals, resulting in a reduction in the amount of data within each subinterval, thus reducing the accuracy of the statistical analysis performed by the collector. How to mitigate this effect is also one of the challenges to be addressed.

In addition, stochastic gradient descent is a common method used in machine learning to find the model parameters. In the stochastic gradient descent process, the gradient needs to be calculated based on the user’s privacy data for each iteration update of the model parameters. In order to protect user privacy, the idea of “local differential privacy” can be used to protect the user’s data in this process. Specifically, after the user calculates the gradient locally, the gradient is perturbed using the privacy-preserving method for numerical data, after which the processed gradient is sent to the data collector for parameter updates. The literature [15,16] illustrates stochastic gradient descent methods in local differential privacy and demonstrates that stochastic gradient descent in small batches can yield better model parameters compared to ordinary stochastic gradient descent methods.

The main contributions of this paper are in the following five areas:(1)A locally hierarchical perturbation method is proposed, an LHP (locally hierarchical perturb) algorithm, for numerical data. This method not only solves the problem that the privacy level needs to be protected, but it also ensures the requirement that the output value domain is the same when different data are perturbed locally;(2)A privacy level conversion method, a PLC (privacy level convert) algorithm, is proposed to increase the amount of available data for each privacy level and thus improve the accuracy of mean estimation, which solves the problem of the reduced statistical accuracy of data caused by data hierarchy;(3)Based on the LHP and PLC algorithms, a hierarchical aggregation method, a HierA algorithm, is proposed for numerical data under local differential privacy, which achieves the hierarchical collection of privacy data and improves data availability while ensuring that users’ privacy needs are satisfied;(4)The proposed hierarchical collection method was applied to small-batch stochastic gradient descent to complete a linear regression task and obtain more accurate prediction models while protecting user privacy;(5)Experimental comparisons with other existing methods on real and synthetic datasets with different distributions were conducted to demonstrate that the proposed method has better usability than the existing methods.

## 2. Related Work

To address the reliance on the data collector’s trustworthiness in the classical centralized differential privacy model, Duchi et al. proposed the local differential privacy model [5]. Since then, numerous scholars have studied the collection of different data types and different statistical analysis tasks for data in the local differential privacy setting.

In the field of frequency estimation for categorical data, the random response (RR) mechanism [17] is the basic perturbation mechanism in the local differential privacy model, which focuses on binary values; i.e., the cardinality of the value domain is 2. The generalized random response (GRR) mechanism [11] improves the RR mechanism by expanding the number of values that can be perturbed to k (k ≥ 2). However, when k is a very large value, the probability that the perturbed data is still the true value is very small, which leads to low availability of the perturbed data. To address this issue, the concept of unary encoding (UE) [12] is proposed, in which privacy data are first encoded into a vector of a specific length before being perturbed. The subsequent perturbation operations act on this vector, and mechanisms such as RAPPOR [9] and OUE [12] based on the concept of UE are proposed one after another.

For the mean estimation problem of numerical data, the Laplace mechanism [2] commonly used in centralized differential privacy can be directly applied to numerical data. However, this approach adds additional noise to each attribute of each datum in a dataset, resulting in low utility of the perturbed data. Duchi et al. [14] proposed their own solution by applying the RR mechanism to the mean estimation of numerical data. The method not only satisfies the local differential privacy requirement but also has an asymptotic error bound. Nguyên et al. [15] proposed the harmony method based on the solution by Duchi et al. The harmony method has the same privacy guarantees and asymptotic error bounds as the method by Duchi et al. while simplifying the data perturbation operation, especially when dealing with multidimensional data. The output value domain after data perturbation in Duchi’s method and the harmony method has only two numbers whose absolute values are greater than 1, so the variance of the perturbed data is always greater than 1 [16]. Even with larger privacy budgets, the accuracy of statistical analysis remains poor. To solve this problem, the piecewise mechanism (PM) was proposed by Wang et al. [16]. The range of values taken from the perturbed data in this method is a continuous interval and the larger the privacy budget, the greater the probability that the output value will be near the true value. Compared with the method by Duchi et al., the accuracy of the PM method is much higher when the privacy budget is large.

In addition, some local differential privacy models for complex data types and specific application scenarios have been proposed one after another. For example, these have included: the data perturbation methods for key-value pair data, PrivKV [22], PCKV [23] and LRR_KV [24]; the data perturbation method for missing data, BiSample [25]; and the personalized models that allow users to set their own privacy budgets, PENA [19], PUM [20] and PLDP [21], etc.

In the application scenario of this paper, the processing of privacy data belongs to the privacy processing of numerical data and the processing of the privacy level belongs to the privacy processing of categorical data.

## 3. Preliminaries and Problem Definition

### 3.1. Preliminaries

Unlike classical centralized differential privacy, localized differential privacy data perturbation is performed locally by the user. The user only sends the perturbed data to the collector, which ensures that no attacker can access the real data other than the user themself. At the same time, the user’s local data perturbation ensures that the attacker cannot infer the user’s real data with high enough probability after obtaining the perturbed data; thus, local differential privacy solves the problem that the data collector must be trusted in the centralized differential privacy model. Local differential privacy is specifically defined as follows.

**Definition 1.** *Local Differential Privacy* [5].

For a randomized perturbation algorithm *M,* it is said to satisfy local differential privacy when it satisfies the following conditions:$$\frac{P_{r}(M(x)=y)}{P_{r}(M(x^{\prime })=y)}\leq e^{\varepsilon }$$
where *x* and $x^{\prime }$ represent any two different inputs to the algorithm *M,* and *y* represents the corresponding output. The smaller the value of *ε* representing the privacy budget, the more difficult it is to identify the corresponding input for the same output, which means that the algorithm has a higher degree of privacy protection.

**Theorem 1.** 
*Sequential Composition.*


For *k* randomized algorithms $M_{i}(1\leq i\leq k)$, if *M_i_*satisfies *ε_i_—*local differential privacy—then the sequence combination of these k randomized algorithms $\left(M_{1},M_{2},\ldots ,M_{k}\right)$ satisfies $\left(\sum_{i=1}^{k}\varepsilon_{i}\right)$—local differential privacy.

Based on this property, the given total privacy budget *ε* can be divided into several parts, each corresponding to a randomized algorithm, so that the raw data can be collected using a sequence of algorithms.

The literature [18] proposes a definition of categorical local differential privacy (ID-LDP) for categorical data whose privacy level needs tend to be different for different categories. Similarly, there is graded local differential privacy for different intervals of similar data, defined as follows.

**Definition 2.** 
*Graded local differential privacy.*


Given a set of privacy budgets $\varepsilon =\{\varepsilon_{1},\varepsilon_{2},\ldots ,\varepsilon_{k}\}$ and a set of data subintervals corresponding to them $D=\{D_{1},D_{2},\ldots ,D_{k}\}$, a perturbation algorithm *M* is said to satisfy graded local differential privacy when it satisfies the following conditions:$$\frac{P_{r}(M(x)=y)}{P_{r}(M(x^{\prime })=y)}\leq e^{r(\varepsilon_{x},\varepsilon_{x^{\prime }})}$$
where $x,x^{\prime }\in U$ represent any two different inputs, $\varepsilon_{x}$ and $\varepsilon_{x^{\prime }}$ represent the privacy budget determined according to the subintervals in which *x* and $x^{\prime }$ are located, respectively, *y* represents the output that the user sends to the collection side, and $r(\varepsilon_{x},\varepsilon_{x^{\prime }})$ represents the privacy budget function with respect to *x* and $x^{\prime }$. For any two different inputs, the degree of indistinguishability of the output is determined jointly by the privacy budgets of the two input data.

Under local differential privacy, the methods for collecting numerical data are generally divided into two categories: one is the Laplace mechanism [2], which is a noise-added method, and the other is the RR mechanism [17], which is a perturbation method, where the user’s continuous numerical data are first discretized according to a certain rule and then perturbed using a specific mechanism to meet the local differential privacy requirements.

The Laplace mechanism [2] is a frequently used method under the classical centralized differential privacy model. Its essence is to add noise to the user’s data that fits the Laplace distribution. For the Laplace distribution Lap(λ), its probability density function is $pdf(x)=\frac{1}{2}e^{\left(-\frac{\left|x\right|}{\lambda }\right)}$, its expectation is 0 and its variance is $2\lambda^{2}$. For all inputs, the output range of the Laplace mechanism is (−∞,+∞) and its expectation of adding noise to all levels of privacy data is 0. Therefore, the simple graded Laplace mechanism solves challenges (1) and (2) and can be used for the graded collection of numerical data. For user data *x* (whose privacy budget is $\varepsilon_{x}$), noise that fits the $Lap(2/\varepsilon_{x})$ distribution is added to it according to their privacy budget, and the user sends the noise-added data to the collector, which directly performs analyses such as mean estimation based on the received data. Since the collection side cannot identify user data of different privacy levels from the collected data, it cannot solve the problem of reduced accuracy of mean estimation due to the small amount of data within each privacy level. In the experimental part, by comparing the analysis on different datasets, it is demonstrated that our proposed HierA method is more advantageous than the graded application of Laplace mechanism.

The GRR method [11] is a local differential privacy perturbation method for categorical data. In this method, for any input x∈{1,2,…,k}, the probability that its output y=x is $p=e^{\varepsilon }/(e^{\varepsilon }+k-1)$ and the probability that y≠x is $q=1/(e^{\varepsilon }+k-1)$.

The harmony method (see Algorithm 1) is a method proposed in the literature [15] for the collection and analysis of numerical type data under local differential privacy, which consists of three main steps: discretization, perturbation and calibration.

The method essentially perturbs the user’s numerical data *v* with a certain probability into discrete $v^{*}$, $v^{*}\in \left\{\frac{e^{\varepsilon }+1}{e^{\varepsilon }-1},-\frac{e^{\varepsilon }+1}{e^{\varepsilon }-1}\right\}$.


**Algorithm 1.** Harmony [15]

**Input:** user’s numerical data *v*, v∈[−1,1] and their privacy budget *ε*

**Output:** perturbed data *v^*^*

1.   Discretize

$v^{*}=\left\{\begin{array}{c} -1\;\;\;\;\;w.p.\;\;\;\;\frac{1-v}{2}\\ 1\;\;\;\;\;\;w.p.\;\;\;\;\frac{1+v}{2} \end{array}\right.$;

2.   Perturb

$v^{*}=\left\{\begin{array}{c} v^{*}\;\;\;\;\;\;w.p.\;\;\;\;\frac{e^{\varepsilon }}{e^{\varepsilon }+1}\\ -v^{*}\;\;\;\;\;w.p.\;\;\;\;\frac{1}{e^{\varepsilon }+1} \end{array}\right.$;

3.   Calibrate

$v^{*}=v^{*}\cdot \frac{e^{\varepsilon }+1}{e^{\varepsilon }-1}$;

4.   Return $v^{*}$


The PM method [16] (see Algorithm 2) is another perturbation method for numerical type data under local differential privacy, which uses a segmented perturbation mechanism. There is a higher probability of perturbing the user data to values in the middle segment of the value domain and a lower probability, to values at both ends.


**Algorithm 2.** Piecewise Mechanism [16]

**Input:** user’s numerical data *v*, v∈[−1,1] and their privacy budget *ε*

**Output:** perturbed data $v^{*}\in [-C,C]$

1.  Sample x uniformly at random from [0,1];

2.  If $X<\frac{e^{\varepsilon /2}}{\left(e^{\varepsilon /2}+1\right)}$:

Sample $v^{*}$ uniformly at random from [l(v),r(v)];

3.  Otherwise:

Sample $v^{*}$ uniformly at random from [(−C,l(v))∪(r(v),C)];

4.  Return $v^{*}$


Where $C=\frac{\left(e^{\varepsilon /2}+1\right)}{\left(e^{\varepsilon /2}+1\right)}$*,*
$l(v_{i})=\frac{C+1}{2}\times v_{i}-\frac{C-1}{2}$, $r(v_{i})=l(v_{i})+C-1$.

### 3.2. Problem Definition

This paper focuses on the hierarchical collection method for numerical data and uses the method for mean estimation.

For the convenience of the study, it is assumed that the user data to be collected takes values in the range [−1,1] and the privacy level is t∈{1,2,…,k} with a privacy budget of $\varepsilon_{t}\in \{\varepsilon_{1},\varepsilon_{2},\ldots ,\varepsilon_{k}\}$ for each level, corresponding to the privacy level. The various symbols used in this paper are shown in Table 1.

## 4. Hierarchical Aggregation for Numerical Data

To address various problems and challenges in the hierarchical collection of numerical data in the local differential privacy environment, the HierA method is proposed in this paper. First, the user perturbs the user’s privacy level and privacy data sequentially and locally using the local hierarchical perturbation method (Algorithm 3). After that, the user sends the privacy level together with the privacy data to the collector, and the collector uses the received user privacy level to classify the user privacy data, first using the privacy level conversion method (Algorithm 4) to process the collected data and then performing a mean estimation from the processed data.

### 4.1. Local Hierarchical Perturbation Mechanism on the User Side

The user’s privacy data *v* are numerical data and privacy levels *t* are categorical data. The privacy data are first queried according to the interval range in which they are located and the privacy levels are perturbed using the GRR method [11], after which the privacy data are discretized and perturbed. The processing of privacy data is borrowed from the harmony mechanism [15], with the difference in the calibration after perturbation placed uniformly on the data collection side in order to reduce the computational overhead locally for the user.

Without the loss of generality, this paper assumes that user privacy data *v* are within [−1,1]. If *v* is not within [−1,1] in the actual scene, the following conversion can be performed using the following conversion rules $v^{\prime }=2\cdot \frac{v-L}{U-L}-1$ [25], where *v* is the original data, which takes value in the range [*L*, *U*], and *v’* is the output data, which takes values in the range [−1,1].

After the user perturbs the privacy level *t* and privacy data *v* locally, the perturbed *<t*, v*>* is sent to the data collector. It is assumed that the privacy level of each subinterval of the user data value domain is sorted from low to high, and the corresponding privacy budgets in each privacy budget set are sequentially decreasing; i.e., $\varepsilon_{1}>\varepsilon_{2}>\ldots >\varepsilon_{k}$. If the privacy level of each subinterval of the user data value domain is not sequentially increasing, then the position of each subinterval can be adjusted so that its privacy level is sequentially increasing. The specific algorithm is as follows.


**Algorithm 3.** Local hierarchical perturbation (LHP)

**Input:** the user’s privacy data $v_{i}\in [-1,1]$, the set of subintervals of the data value field $D=\{D_{1},D_{2},\ldots ,D_{k}\}$ and the set of privacy budgets corresponding to each subinterval $\varepsilon =\{\varepsilon_{1},\varepsilon_{2},\ldots ,\varepsilon_{k}\}$

**Output:** perturbed tuple *<t_i_*, v_i_*>*

1.  According to the interval in which *v*_*i*_ is located, find out its corresponding pri-vacy level $t_{i},\;t_{i}\in \{1,2,\ldots ,k\}$;

2.  Perturb *t_i_*:



$P_{r}(t_{i}^{*}=d)=\left\{\begin{array}{c} \frac{e^{\varepsilon_{t_{i}}}}{e^{\varepsilon_{t_{i}}}+k-1}\;\;\;\;if\;\;\;d=t_{i}\\ \frac{1}{e^{\varepsilon_{t_{i}}}+k-1}\;\;\;if\;\;\;d\neq t_{i} \end{array}\right.$



where $d,t_{i}\in \{1,2,\ldots ,k\}$;

3.   Discretize *v_i_*:

$v_{i}^{*}=\left\{\begin{array}{c} -1\;\;\;\;\;w.p.\;\;\;\;\frac{1-v_{i}}{2}\\ 1\;\;\;\;\;\;w.p.\;\;\;\;\frac{1+v_{i}}{2} \end{array}\right.$;

4.   Perturb *v_i_^*^*:

$v_{i}^{*}=\left\{\begin{array}{c} \;\;\;v^{*}\;\;\;\;\;\;w.p.\;\;\;\;\;\;\frac{e^{\varepsilon_{t_{i}^{*}}}}{e^{\varepsilon_{t_{i}^{*}}}+1}\\ -v^{*}\;\;\;\;\;\;w.p.\;\;\;\;\;\;\frac{1}{e^{\varepsilon_{t_{i}^{*}}}+1} \end{array}\right.$;

5.   Obtain the perturbed tuple *<t_i_*, v_i_*>*.


### 4.2. Calibration Analysis at the Collection End

We then needed to address the problem that the amount of data at each level decreases after data hierarchical collection, resulting in larger mean estimation errors. In this paper, we propose a privacy level transformation method (PLC algorithm) to significantly increase the amount of available data within each privacy level (especially within the high privacy level), which leads to a significant improvement in the accuracy of the mean value estimation.

Suppose there is user data *v* after graded perturbation v∈{−1,1}, whose privacy level is *i*. Its corresponding privacy budget is *ε_i,_* to which a random perturbation has been applied using the random response mechanism. Then, its perturbation probability, i.e., the probability of remaining unchanged, is $p_{i}=\frac{e^{\varepsilon_{i}}}{e^{\varepsilon_{i}}+1}$. Now, we want to perturb it for the second time so that it has privacy level *j,* privacy budget *ε_j_* and perturbation probability *p_j,_* $p_{j}=\frac{e^{\varepsilon_{j}}}{e^{\varepsilon_{j}}+1}$. Firstly, assume that the probability of the second perturbation is *q*, then *q* should satisfy $p_{i}\cdot q+(1-p_{i})(1-q)=p_{j}$. We can obtain $q=\frac{p_{i}+p_{j}-1}{2p_{i}-1}$. Accordingly, the probability that *v* performs a flip in the second perturbation is $1-q=\frac{p_{i}+p_{j}-1}{2p_{i}-1}$. The overall flip probability is $p_{i}\cdot \frac{p_{i}-p_{j}}{2p_{i}-1}+(1-p_{i})\frac{p_{i}+p_{j}-1}{2\cdot p_{i}-1}=1-p_{j}$. Therefore, using $\frac{p_{i}+p_{j}-1}{2p_{i}-1}$ as the perturbation probability of the second perturbation can perturb *v* into the data with privacy level j. Note that we can only perturb low-level data into high-level data; i.e., *p_i_* must be greater than *p_j_*. The specific steps are as follows.

**Algorithm 4.** Privacy level conversion (PLC)
**Input:** the dataset *V_i_* with privacy level *i*, the number of times of data reuse *μ* and the set of privacy budgets $\varepsilon =\{\varepsilon_{1},\varepsilon_{2},\ldots ,\varepsilon_{n}\}$

**Output:** the set of converted datasets $\left\{V_{i},V_{i(i+1)},V_{i(i+2)},\ldots ,V_{i(min(k,i+\mu -1))}\right\}$

1.   For j = i + 1 to min(k,i+μ−1)

2.   For *v* in *V_i_*:

3.  $v=\left\{\begin{array}{l} v\;\;\;\;\;w.p.\;\;\;\;\;\frac{p_{i}+p_{j}-1}{2p_{i}-1}\\ -v\;\;\;\;w.p.\;\;\;\;\;\frac{p_{i}-p_{j}}{2p_{i}-1} \end{array}\right.$

where $p_{i}=\frac{e^{\varepsilon_{i}}}{e^{\varepsilon_{i}}+1}$, $p_{j}=\frac{e^{\varepsilon_{j}}}{e^{\varepsilon_{j}}+1}$;

4.   Obtain $\left\{V_{i},V_{i(i+1)},V_{i(i+2)},\ldots ,V_{i(min(k,i+\mu -1))}\right\}$.


The reuse of privacy data is achieved by converting data with a lower privacy level to data with a higher privacy level. The number of rank conversions varies depending on the number of reuses set by the system. Theoretically, it is possible to convert low-level data to any high-level data, but the computational overhead will increase as the number of conversions increases.

The HierA algorithm for the hierarchical collection of numerical data under local differential privacy is given by combining Algorithms 3 and 4, and the specific steps are shown in Algorithm 5.


**Algorithm 5.** Hierarchical aggregation for numerical data

**Input:** the users’dataset $V=\{v_{1},v_{2},\ldots ,v_{n}\}$ the set of subintervals of the data value field $D=\{D_{1},D_{2},\ldots ,D_{k}\}$, the set of privacy budgets $\varepsilon =\{\varepsilon_{1},\varepsilon_{2},\ldots ,\varepsilon_{n}\}$ and the number of data reuse μ∈{1,2,…,k}

**Output:** estimated mean $m^{*}$

User side:

1.  The user perturbs their data $v_{i}$ locally using the LHP method to obtain the perturbed tuple $<t_{i}^{*},v_{i}^{*}>$:

$<t_{i}^{*},v_{i}^{*}>=LHP(D,\varepsilon ,v_{i})$;

Collection side:

2.  The received user data are classified according to the privacy level to obtain b $\left\{V_{1},V_{2},\ldots ,V_{k}\right\}$;

3.  The classified dataset is transformed using the PLC algorithm to obtain the transformed dataset:

  $\;\{V_{i},V_{i(i+1)},V_{i(i+2)},\ldots ,V_{i(min(k,i+\mu -1))}\}=PLC(V_{i},\varepsilon ,\mu )$

4.  The following dataset matrix is obtained, with each column representing a privacy level:



$\left[\begin{array}{cccccc} V_{1} & V_{12} & \cdots & V_{1(min(k,\mu ))} & & \\ & V_{2} & V_{23} & \cdots & V_{2(min(k,\mu +1))} & \\ & & \ddots & \ddots & \ddots & \ddots \end{array}\right]$



5.  The datasets with the same privacy level are merged and the compensation dataset is added:

$\mathrm{for}\;i\;\mathrm{in}\;\left\{1,2,\ldots ,k\right\}$:

if i+μ−1>k:



$V_{i}^{*}=(\mu -k+i)V_{i}$



if i<μ:

     $V_{i}^{*}=V_{i}+V_{(i-1)i}+\ldots +V_{1i}$

     otherwise μ<=i<k−μ+2:

        $V_{i}^{*}=V_{i}+V_{(i-1)i}+\ldots +V_{(i-\mu +1)i}$

Obtain $V^{*}=\{V_{1}^{*},V_{2}^{*},\ldots ,V_{k}^{*}\}$;

6.  Perform the following for each dataset $V_{i}^{*}$ in $\;\mathrm{in}\;V^{*}$:

The number of 1 and −1 in $V_{i}^{*}$ is denoted as *n_1_* and *n_2_*, respectively:



$N_{i}=n_{1}+n_{2}$


$,\;p_{i}=\frac{e^{\varepsilon_{i}}}{e^{\varepsilon_{i}}+1}$



$n_{1}^{*}=\frac{p_{i}\cdot N-n_{2}}{2p_{i}-1}$;

$n_{2}^{*}=\frac{p_{i}\cdot N-n_{1}}{2p_{i}-1}$;

Correct $n_{1}^{*}$ and $n_{2}^{*}$ by making them equal to N if they are greater than N or equal to 0 if they are less than 0.

$S(V_{i}^{*})=n_{1}^{*}-n_{2}^{*}$;

7.  Calculate the estimated mean:$m^{*}=\frac{1}{\left|V^{*}\right|}\sum_{i=1}^{k}S(V_{i}^{*})$.


In step 3, when i+μ−1>k, the number of transformations of dataset $V_{i}$ may be less than μ−1. To ensure that the final mean estimate is unbiased, it is necessary to add μ−k+i datasets $V_{i}$ as the compensation dataset in step 5.

### 4.3. Privacy and Usability Analysis

**Theorem 2.** 
*Algorithm 3 satisfies the graded local differential privacy requirement.*


In the application scenario of the proposed method, the user’s original data and the adopted privacy level are both private data, where the privacy level is determined by the user’s original data. Therefore, both the perturbation of the user’s privacy level and the perturbation of the user’s privacy data need to satisfy the local differential privacy requirements.

Perturbation for user privacy level *t_i_*:


$$\begin{array}{l} \frac{P_{r}(t^{*}|t_{1})}{P_{r}(t^{*}|t_{2})}\leq \frac{P_{r}(t^{*}=t_{1}|t_{1})}{P_{r}(t^{*}=t_{1}|t_{2})}\\ \;\;\;\;\;\;=\frac{\frac{e^{\varepsilon_{t_{1}}}}{e^{\varepsilon_{t_{1}}}+k-1}}{\frac{1}{e^{\varepsilon_{t_{2}}}+k-1}}\leq e^{\max(\varepsilon_{t_{1}},\varepsilon_{t_{2}})} \end{array}$$


Thus, the processing of user privacy level *t_i_* satisfies the hierarchical local differential privacy requirements.

2.Perturbation for user privacy data *v_i_*:

The perturbation of *v_i_* is related to the perturbation of *t_i_*, the privacy levels after the perturbation of *t_1_* and *t_2_* are $t_{1}^{*}$ and $t_{2}^{*}$, respectively, and the perturbed output values have only two values (−1 and 1), which we consider the case $v^{*}=1$ (similarly for $v^{*}=-1$).
$$\begin{array}{l} \frac{P_{r}(v^{*}=1|v_{1})}{P_{r}(v^{*}=1|v_{2})}=\frac{\frac{1+v_{1}}{2}\cdot \frac{e^{\varepsilon_{t_{1}^{*}}}}{e^{\varepsilon_{t_{1}^{*}}}+1}+\frac{1-v_{1}}{2}\cdot \frac{1}{e^{\varepsilon_{t_{1}^{*}}}+1}}{\frac{1+v_{2}}{2}\cdot \frac{e^{\varepsilon_{t_{2}^{*}}}}{e^{\varepsilon_{t_{2}^{*}}}+1}+\frac{1-v_{2}}{2}\cdot \frac{1}{e^{\varepsilon_{t_{2}^{*}}}+1}}\\ \;\;\;\;\;\;\;\;\;=\frac{e^{\varepsilon_{t_{2}^{*}}}+1}{e^{\varepsilon_{t_{1}^{*}}}+1}\cdot \frac{v_{1}(e^{\varepsilon_{t_{1}^{*}}}-1)+e^{\varepsilon_{t_{1}^{*}}}+1}{v_{2}(e^{\varepsilon_{t_{2}^{*}}}-1)+e^{\varepsilon_{t_{2}^{*}}}+1}\\ \;\;\;\;\;\;\;\;\;\;\;\;\;\;\;\;\;\leq \frac{e^{\varepsilon_{t_{2}^{*}}}+1}{e^{\varepsilon_{t_{1}^{*}}}+1}\cdot e^{\varepsilon_{t_{1}^{*}}}\\ \;\;\;\;\;\;\;\;\;\;\;\;\;\;\;\;\;\leq e^{\max(\varepsilon_{t_{1}^{*}},\varepsilon_{t_{2}^{*}})} \end{array}$$

Therefore, the processing of user data vi satisfies the hierarchical local differential privacy requirements.

**Theorem 3.** *The mean estimate obtained by Algorithm 5 is an unbiased estimate*.

**Proof of Theorem 3.** By denoting the true mean of the user dataset V as $m=\frac{1}{n}\sum_{i}^{n}v_{i}$ and the mean after discretizing the user data in Algorithm 3 as $m^{\prime }$, we have:
$$\begin{array}{l} E(m^{\prime })=E(\frac{1}{n}\sum_{i}^{n}v_{i}^{*})\\ \;\;\;\;\;\;=\frac{1}{n}\sum_{i}^{n}\left(-1\cdot \frac{1-v_{i}}{2}+\frac{1+v_{i}}{2}\right)\\ \;\;\;\;\;\;=\frac{1}{n}\sum_{i}^{n}v_{i}\\ \;\;\;\;\;\;=m \end{array}$$Therefore, the mean estimation of the user discretized data is unbiased. The random perturbation of the discretized data in Algorithm 3 causes bias, so the user data is corrected in Algorithm 5.Step 6 in Algorithm 5 corrects the data after user discretization. For any privacy level dataset $V_{i}^{*}$, denote its true mean as $m_{i}$. Denote the number of true 1 and −1 in the user dataset before the local perturbation as ${\overset{⌢}{n}}_{1}$ and ${\overset{⌢}{n}}_{2}$, the number of 1 and −1 received by the collector after the perturbation as $n_{1}$ and $n_{2}$ and the number of 1 and −1 corrected by the collector as $n_{1}^{*}$ and $n_{2}^{*}$. So, we have:
$$E(m_{i}^{*})=\frac{{\overset{⌢}{n}}_{1}+{\overset{⌢}{n}}_{2}}{{\overset{⌢}{n}}_{1}+{\overset{⌢}{n}}_{2}}=m_{i}$$
$$\begin{array}{l} E(n_{1}+n_{2})=p_{i}{\overset{⌢}{n}}_{1}+(1-p_{i}){\overset{⌢}{n}}_{2}+p_{i}{\overset{⌢}{n}}_{2}+(1-p_{i}){\overset{⌢}{n}}_{1}\\ \;\;\;\;\;\;\;\;\;\;\;={\overset{⌢}{n}}_{1}+{\overset{⌢}{n}}_{2} \end{array}\;\begin{array}{l} E(n_{1}^{*})=E(\frac{p_{i}-1}{2p_{i}-1}\cdot N_{i}+\frac{n_{1}}{2p_{i}-1})\\ \;\;\;\;\;\;=\frac{p_{i}-1}{2p_{i}-1}E(n_{1}+n_{2})+\frac{E(n_{1})}{2p_{i}-1}\\ \;\;\;\;\;\;=\frac{(p_{i}-1)({\overset{⌢}{n}}_{1}+{\overset{⌢}{n}}_{2})+p_{i}{\overset{⌢}{n}}_{1}+(1-p_{i}){\overset{⌢}{n}}_{2}}{2p_{i}-1}\\ \;\;\;\;\;\;={\overset{⌢}{n}}_{1} \end{array}$$Similarly:$$\begin{array}{l} E(n_{1}^{*})=E(\frac{p_{i}-1}{2p_{i}-1}\cdot N_{i}+\frac{n_{2}}{2p_{i}-1})\\ \;\;\;\;\;\;={\overset{⌢}{n}}_{2} \end{array}$$Therefore, for dataset $V_{i}^{*}$, its estimated mean $m_{i}^{*}$ satisfies:$$E(m_{i}^{*})=E(\frac{n_{1}^{*}-n_{2}^{*}}{n_{1}^{*}+n_{2}^{*}})=\frac{{\overset{⌢}{n}}_{1}-{\overset{⌢}{n}}_{2}}{{\overset{⌢}{n}}_{1}+{\overset{⌢}{n}}_{2}}=m_{i}$$Noting the estimated mean value in Algorithm 5 as $m^{*}$, we have:$$\begin{array}{l} E(m^{*})=E(\frac{1}{\left|V^{*}\right|}\sum_{i=1}^{k}(N_{i}\cdot m_{i}^{*}))\;\\ \;\;\;\;\;\;\;=\frac{1}{\left|V^{*}\right|}\sum_{i=1}^{k}(N_{i}\cdot m_{i}))\;\\ \;\;\;\;\;\;\;=m \end{array}$$Therefore, the mean estimate in Algorithm 5 is an unbiased estimate. □

## 5. Application of Hierarchical Aggregation in Stochastic Gradient Descent

Stochastic gradient descent is a common method for finding model parameters in machine learning. The linear regression model is used as an example to study the application of hierarchical aggregation method in stochastic gradient descent.

Assume that each user has a set of multidimensional data $<x_{i},y_{i}>$, $x_{i}\in {\left[-1,1\right]}^{d}$, $y_{i}\in [-1,1]$. The target model is a linear model $f(x_{i})=$$\alpha^{T}x_{i}+b$. Let $\beta =(b,\alpha_{1},\alpha_{2},\ldots ,\alpha_{d})$, such that the final goal of model training is to obtain a parameter vector $\beta^{*}$, satisfying the following condition:
$$\beta^{*}=\underset{\beta }{arg\;\;min}[\frac{1}{n}\sum_{i=1}^{n}L(\beta ;x_{i},y_{i})+\frac{\lambda }{2}{\left\|\beta \right\|}_{2}^{2}]$$
where L(⋅) denotes the loss function $L(\beta ;x_{i},y_{i})={\left(\beta^{T}x_{i}-y_{i}\right)}^{2}$, $\frac{\lambda }{2}{\left\|\beta \right\|}_{2}^{2}$ denotes the regular term and λ denotes the regular term coefficient.

The literature [9,10] has demonstrated that in a local differential privacy setting, the use of small-batch stochastic gradient descent can yield a more accurate prediction model than the ordinary stochastic gradient descent method. Specifically, a parameter vector $\beta_{0}^{*}$ is first initialized and then iteratively updated by the following equation:$$\beta_{t+1}=\beta_{t}-\gamma_{t}\cdot \frac{1}{\left|G\right|}\sum_{i=1}^{\left|G\right|}\nabla R(\beta_{t};x_{i},y_{i})$$
where $R(\beta ;x_{i},y_{i})=L(\beta ;x_{i},y_{i})+\frac{\lambda }{2}{\left\|\beta_{t}\right\|}_{2}^{2}$, $\nabla R(\beta_{t};x_{i},y_{i})$ is the gradient of $R(\beta ;x_{i},y_{i})$ at $\beta_{t}$; $\gamma_{t}$ represents the learning efficiency at the *t^th^* iteration $\gamma_{t}=O(1/\sqrt{t})$; and $\left|G\right|$ represents the number of samples used in each iteration, i.e., the number of users per group $\left|G\right|=\Omega (\frac{d\log d}{\varepsilon^{2}})$.

The collector first sends the existing parameter vector $\beta_{t}^{*}$ to a group of $\left|G\right|$ users in each iteration, and each user in the group calculates the gradient ∇R based on the received $\beta_{t}^{*}$ and then uses a perturbation algorithm for numerical data to privatize ∇R and sends it to the collector. After receiving the gradients from a set of users, the collector calculates the mean value for the update of the parameter vector to obtain $\beta_{t+1}^{*}$, and this paper uses Algorithm 4 to implement the privacy processing of ∇R and to find the mean value. It is important to note that in order to achieve the hierarchical protection for user data, the classification of ∇R is not based on the size of each ∇R component itself, but on the privacy classification of certain attribute data corresponding to the ∇R component. Referring to the gradient clipping method commonly used in deep learning, the user calculates ∇R and then corrects each component by making it equal to 1 if it is greater than 1 or equal to −1 if it is less than −1.

In ordinary stochastic gradient descent, each sample datum often needs to participate in multiple iterations until the parameter vector changes are small enough. However, in the local differential privacy setting, if each sample point is involved in multiple iterations, the privacy budget used by the user will be severely divided each time to protect the user’s privacy, resulting in a drastic decrease in the usability of the user’s data. Therefore, in the local differential privacy environment, each user participates in only one cycle.

## 6. Experiments

Experiments were conducted using synthetic datasets conforming to uniform distribution, Gaussian distribution, exponential distribution and several real datasets as the users’ privacy data, respectively, and the proposed method was fully analyzed in a cross-sectional and longitudinal comparison.

The data ranges of the synthetic datasets were all set to [−1,1], with a mean of 0.3 and a standard deviation of 0.2 for the Gaussian distribution dataset and a standard deviation of 0.3 for the exponential distribution dataset.

The real dataset uses the 2010 census data for Brazil and the United States extracted from the Integrated Public Use Microdata Series [26], denoted as the BR dataset and the US dataset, respectively. The BR dataset has approximately 3.86 million records, each containing four numerical and six categorical attributes. The US dataset contains approximately 1.52 million records, each containing five numerical and five categorical attributes.

For the category-type attributes in the dataset, if there is an order relationship between the attribute values, they are converted into consecutive values. For example, the two attribute values “Does not speak English” and “Speaks English“ are converted to {0,1}. If there is no sequential relationship between the attribute values, they are converted to k attributes according to the k attribute values they have; for example, the attribute values “male” and “female” are converted to (1,0) and (0,1). After that, the range of all the data is converted to [−1,1].

The interval division of user data and the privacy budget allocation of each privacy level should be set according to the specific needs of the actual problem, and the specific setting method is out of the scope of this paper. In the following experiments, five privacy levels are used as an example, and the value domain is divided into five subintervals equally according to the size of data values {[−1,−0.6),[−0.6,−0.2),…,[0.6,1.0]} and the set of privacy budgets ε={5ε,4ε,3ε,2ε,ε}.

The mean absolute error (*MAE*) as well as the mean square error (*MSE*) were used to evaluate the utility of the numerical data collection methods in performing mean estimation. The calculation equations were:$$MAE(m^{*})=\frac{1}{T}\sum \left|m^{*}-m\right|$$
$$MSE(m^{*})=\frac{1}{T}\sum {\left(m^{*}-m\right)}^{2}$$
where *m* represents the true mean of the user data, $m^{*}$ represents the estimated mean and *T* represents the number of experiments.

### 6.1. Different Data Reuse Times μ

The “age” attribute was extracted from the US dataset as the experimental dataset (US_AGE dataset), and the number of data reused *µ* was varied to observe the accuracy of mean estimation using the HierA method, using MAE as the evaluation metric *T* = 100.

The size of the data reuse count *µ* determines the number of rank conversions of low ranked data for mean estimation on the collection side. The larger the value of *µ*, the greater the number of conversions, and the greater the amount of data used for mean estimation, the greater the computational overhead. The range of *µ* is {1,2,...,*k*} without considering computational overhead where *µ* = 1 corresponds to no rank conversion and *µ* = *k* means converting a rank datum to any higher ranked datum.

Figure 1 shows that the higher the value of *µ*, the higher the accuracy of the mean estimation. This is mainly due to the fact that the larger the value of *µ*, the larger the amount of data available for each privacy level and therefore the more accurate the mean estimation.

### 6.2. Comparison of Different Methods

The HierA method (*µ* = 2) proposed in this paper is compared with the harmony method, PM method and graded Laplace method for MAE when performing mean estimation on synthetic and real datasets with different distributions. Note that the HierA method proposed in this paper is a graded data collection method. The privacy level of user data may vary depending on its size. The harmony and PM methods are single privacy-level methods, and to ensure users’ privacy security, all users in these two methods must use the highest privacy level, i.e., the smallest privacy budget at the time of perturbation. Furthermore, the graded Laplace method uses the same graded settings as the HierA method. The experiments compared the MAE of the different methods when data collection was performed and mean estimation was performed under different privacy budgets ε.

From Figure 2, we can see that the HierA method proposed in this paper for the hierarchical collection of numerical data has a higher accuracy than the existing methods on different datasets, especially when the privacy budget *ε* is small, with the advantage of the HierA method being more obvious. This is mainly due to the fact that the proposed method not only assigns appropriate privacy levels to different privacy data according to their own characteristics to avoid the unnecessary loss of data availability, but also expands the amount of data available at each level for mean estimation by transforming and merging the privacy levels of data, which solves the problem of data reduction at each level due to data grading.

### 6.3. Small-Batch Stochastic Gradient Descent

Perform small-batch stochastic gradient descent experiments on the BR and US datasets. Perform small-batch stochastic gradient descent experiments using the processed BR and US datasets to complete the linear regression task. Specifically, the “total income” attribute is used as output *y* and the other attributes are used as input ***x***. According to [9], the parameters are set as follows: the number of people per group $G=\frac{d\log d}{\varepsilon^{2}}$, learning efficiency $\gamma_{t}=(1/\sqrt{t})$ and the penalty term coefficient $\lambda =10^{-4}$.

Privacy-preserving treatments were applied to the gradients of each set of user data using the harmony, PM, graded Laplace and HierA methods proposed in this paper. The MSE with 10 times of 5-fold cross-validation was used to evaluate the prediction accuracy of the obtained linear models when different privacy-preserving methods were used.

As can be seen from Figure 3, in the linear regression task, the accuracy of the prediction models obtained using graded privacy-preserving methods (graded Laplace and HierA method) is higher than that of the models obtained by single-grade privacy-preserving methods. Additionally, the HierA method proposed in this paper is more advantageous than the simple graded Laplace method.

## 7. Conclusions

To address the problem that numerical privacy data of the same type in different ranges require different degrees of privacy protection, this paper proposes the HierA method. This method uses the LHP algorithm to implement a hierarchical perturbation of user data and the PLC algorithm to implement a privacy level conversion of the perturbed data. The method was then applied to a small-batch stochastic gradient descent to complete a linear regression task.

Through theoretical analysis, it is demonstrated that the method satisfies the local differential privacy requirement and that mean estimation is unbiased. Finally, through experiments, it is demonstrated that the hierarchical collection method of numerical data proposed in this paper outperforms the existing local differential privacy-preserving methods in both simple mean estimation and small-batch stochastic gradient descent. The impact of user data distribution on the accuracy of the hierarchical collection method will be further investigated in the future and the optimal hierarchical collection method under different distributions will be proposed.

## Figures and Tables

**Figure 1 sensors-23-01115-f001:**
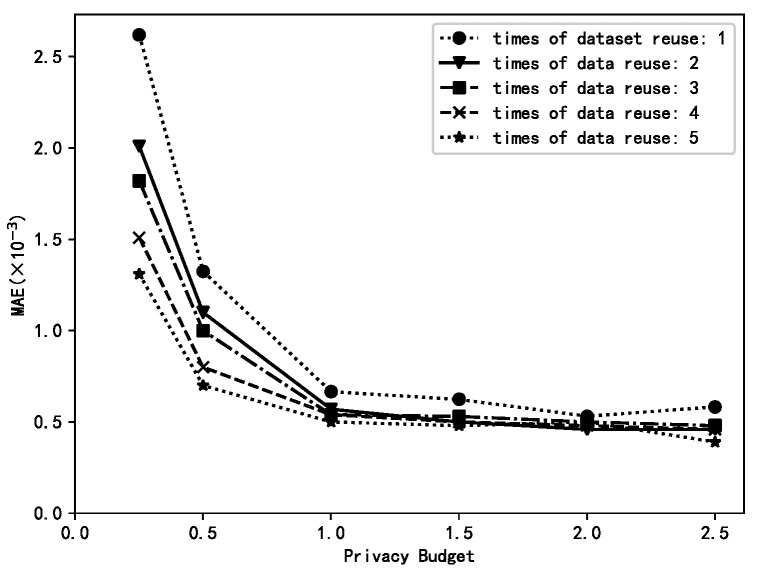
The impacts of different *μ* values.

**Figure 2 sensors-23-01115-f002:**
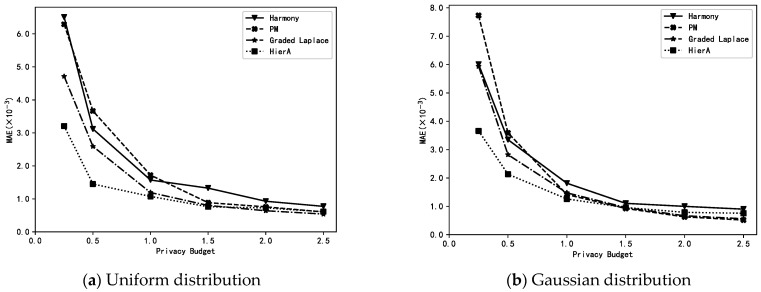
Estimations of the mean using different methods.

**Figure 3 sensors-23-01115-f003:**
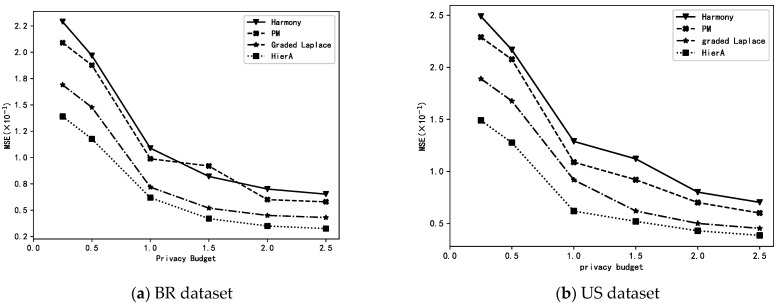
Linear regressions using different methods.

**Table 1 sensors-23-01115-t001:** Symbol Definitions.

Symbol	Description
$U=\{u_{1},u_{2},\ldots ,u_{n}\}$	User set $u_{i}$ indicating the *i*-th user
$V=\{v_{1},v_{2},\ldots ,v_{n}\}$	User’s dataset $v_{i}$ denoting user $u_{i}\prime s$ data
$D=\{D_{1},D_{2},\ldots ,D_{k}\}$	Subintervals of data value fields by privacy level
$\varepsilon =\{\varepsilon_{1},\varepsilon_{2},\ldots ,\varepsilon_{k}\}$	Collection of privacy budgets for user data, with k levels
μ∈{1,2,…,k}	Number of data reused
*v_i_**	User data after perturbation
$t_{i}$	Privacy level of user data $v_{i}$
*t_i_**	Privacy level after perturbation
*p*	Probability of user data remaining unchanged
*m*	True mean of the user’s dataset
m*	Estimated mean of the user’s dataset

## Data Availability

Publicly available datasets were analyzed in this study. This data can be found here: https://www.ipums.org accessed on 7 December 2022.

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
