# Peer review of "Hierarchical Aggregation for Numerical Data under Local Differential Privacy"

_sensors, 2023, doi:10.3390/s23031115_

Round 1

Reviewer 1 Report

1. Some symbols are not clearly defined or explained, and it is better to put Table 1 at the beginning of Section 3.

2. This paper compared the experimental result of the proposed method HierA with some previous method, including Harmony method, PM method and graded Laplace method. Only the Harmony was introduced in algorithm 1 and it was proposed in 2016. Are there any other related works proposed in recent years? If so, the authors should do the comparison with them. If no, the authors should introduce the state of the art works in this field.

3. Why the authors select BR dataset and US dataset? Do these datasets and your program are available for open access?

Author Response

Thank you very much for reviewing our paper. Your comments are very pertinent and necessary. We have explained or revised each of them, and the following are the required explanations. You can see the revisions in the attached manuscript.

  1. I checked the symbols that appeared in the paper and placed all the symbols that were not clearly defined or explained in Table 1.

  1. A detailed description of the PM method is added in Section 3. In addition, a single-grade Laplace mechanism is introduced in Section 3, whereas the graded Laplace method is only a simple grading of the single Laplace mechanism, so no extra space is used in the paper to introduce the specific operations of each step of this method. In addition, in Section 2, we add a description of the latest research work in this field. Given that the PM method, Harmony method, and graded Laplace method are more representative and that the methods proposed by some recent scholars do not have particularly obvious advantages over these methods, we decided to use these three methods and the HierA method proposed in this paper for comparison experiments.

  1. Since the two datasets, BR and US, are more reliable, easy to obtain, and have a large amount of data to support our experimental results, we choose to complete our experiments using these two datasets. These two datasets can be downloaded from https://www.ipums.org. Also, we openly shared our experimental code on GitHub (https://github.com/hao606/HierA).

Thank you again for your work in reviewing our paper.

Reviewer 2 Report

A separate page is attached. 

Author Response

Thank you very much for reviewing our paper. Your comments are very pertinent and necessary. We have explained or revised each of them based on your review comments, and the following are the required explanations. You can see the revisions in the attached manuscript.

  1. Thanks again for your advice. I have read carefully the paper you recommended (https://doi.org/10.3390/s20247030) as well as some other recent related papers. I have really benefited from them. I have revised the introduction and related work section to present more research in this area. Again, I am grateful for your recommendation.
  2. I have checked the sentences in the paper and made changes to make them simpler and easier to read if they are too long or too complicated.
  3. Stochastic gradient descent is one of the most commonly used methods for solving model parameters, and in the process of performing iterative parametrization, each component of the gradient used in each iteration is numerical data, which can be protected using the method proposed in this paper. References [15, 16] both chose to apply their proposed local differential privacy models to stochastic gradient descent. Therefore, stochastic gradient descent is chosen.
  4. Any sentences that were not properly expressed, such as on Page 2, First Paragraph ('. there is little research on.'), have been revised.
  5. I revised the related work to make it more detailed and comprehensive.
  6. I have added citation marks for the mathematics in Section 3 and Section 4.
  7. The conclusion section has been revised to provide a more detailed and comprehensive presentation.

Thank you again for your work in reviewing our papers.

Reviewer 3 Report

 I enjoyed reading your manuscript. Your topic was highly important since it focused on local differential privacy, which solves the problem that the data collector must be trusted in centralized differential privacy models. Your procedures for conducting this study were adequate. Also, I found your careful logic that led to the findings and recommendations.

Author Response

Thank you very much for reviewing our paper. We have made some improvements to the paper. You can see them in the attachment.

Round 2

Reviewer 2 Report

The authors have incorporated the recommended issues. 

Reviewer 3 Report

I am satisfied with the corrections made. Thanks.